# Effect of Vitamin D Supplementation on Muscle Status in Old Patients Recovering from COVID-19 Infection

**DOI:** 10.3390/medicina57101079

**Published:** 2021-10-09

**Authors:** Alberto Caballero-García, Daniel Pérez-Valdecantos, Pablo Guallar, Aurora Caballero-Castillo, Enrique Roche, David C. Noriega, Alfredo Córdova

**Affiliations:** 1Department of Anatomy and Radiology, Faculty of Health Sciences, GIR of Physical Exercise and Aging, University Campus “Los Pajaritos”, Valladolid University, 42004 Soria, Spain; alberto.caballero@uva.es; 2Department of Biochemistry, Molecular Biology and Physiology, Faculty of Health Sciences, GIR of Physical Exercise and Aging, University Campus “Los Pajaritos”, Valladolid University, 42004 Soria, Spain; danielperezvaldecantos@gmail.com; 3SACYL, Soria Norte Health Center, Avda Espolon s/n, 42001 Soria, Spain; pguallar@saludcastillayleon.es (P.G.); aurora95_narros@hotmail.com (A.C.-C.); 4Department of Applied Biology—Nutrition, Institute of Bioengineering, University Miguel Hernández, 03202 Elche, Spain; eroche@umh.es; 5Alicante Institute for Health and Biomedical Research (ISABIAL), 03010 Alicante, Spain; 6CIBER Physiopathology of Obesity and Nutrition (CIBEROBN), Instituto de Salud Carlos III (ISCIII), 28029 Madrid, Spain; 7Department of Surgery, Ophthalmology, Otorhinolaryngology and Physiotherapy, Faculty of Medicine, Hospital Clínico Universitario de Valladolid, 47003 Valladolid, Spain; davidcesar.noriega@uva.es

**Keywords:** biochemistry, blood count, exercise, spirometry, vitamin D

## Abstract

*Background and Objectives*: Vitamin D, in addition to its effect on mineral homeostasis, plays a key role in muscle metabolism. Vitamin D supplementation is involved in muscle recovery after damage as a consequence of either pathology or after high-intensity exercise. In this context, the aim of this study was to analyze the effect of vitamin D on muscle fitness in elderly patients in the recovery phase after SARS-CoV-2 (COVID-19) infection. *Materials and Methods*: This pilot study was conducted at the Soria Norte Health Center. The study consisted of a double-blind trial with two groups of men (placebo and vitamin D-supplemented) (*n* = 15/group). Treatment with vitamin D (cholecalciferol: 2000 IU/day) and placebo was carried out for 6 weeks. Circulating hematological and biochemical parameters (total protein, glucose, vitamin D, urea, uric acid, aspartate aminotransferase/glutamic-oxaloacetic transaminase, alanine aminotransferase/glutamic-pyruvic transaminase, creatine kinase, lactate dehydrogenase, aldolase, gamma-glutamyl transferase and myoglobin) and the hormones cortisol and testosterone were determined. As for respiratory function tests, FEV1 and respiratory flow were also studied. For physical fitness tests, the “six-minute walk test” (6MWT) was used. *Results*: After vitamin D supplementation, we observed that serum creatine kinase levels returned to optimal values. This change suggests a protective role of vitamin D against muscle catabolism compared to placebo. In terms of physical test results, we observed only slight non-significant improvements, although patients reported feeling better. *Conclusions*: Vitamin D supplementation produces decreases in indicators of muscle damage, which may ultimately contribute to improving the health status and quality of life of patients who have suffered from COVID-19, during the recovery process.

## 1. Introduction

The high transmissibility and pathogenicity of SARS-CoV-2 are posing a major challenge to health services. The virus causes generally mild symptoms but, in some cases, produces a severe acute respiratory syndrome in addition to myalgias of neurogenic origin [1,2]. During infection, damaged cells trigger lung inflammation, largely mediated by proinflammatory macrophages and granulocytes, resulting in fever, cough, fibrosis and a dramatic increase in proinflammatory cytokine levels [3].

Cytokines are a group of glycoproteins secreted by cells that regulate different functions of the immune system [3,4]. In acute cases, an exaggerated inflammatory response, termed “hypercytokinemia”, has been described, which will determine the prognosis of COVID-19 infection [5]. Cellular processes modulated by cytokines include activation/anergy/apoptosis, proliferation, differentiation and maturation of lymphocytes, and of accessory cells. These molecules are also involved in the regulation of circulating levels and tissue distribution of leukocytes. Cytokines also exert modulatory effects on cells in various organs and body systems. Functional pleiotropy and redundancy are characteristic of cytokines, making their functional regulation an important preventive and therapeutic target in inflammatory diseases [5,6]. In this context, muscle tissue is a major target of the inflammatory process caused after COVID-19 infection [1,2].

According to what has been reported in the literature, vitamin D is an immunomodulatory micronutrient [7,8]. Several observational studies have established the association between low vitamin D levels and an increased risk of acute respiratory viral infections (ARVI) [9,10,11,12]. Shi et al. [13] observed that low blood levels of vitamin D are associated with acute lung injury and acute respiratory distress syndrome. In this regard, low vitamin D levels have been observed in SARS-CoV-2-positive patients [14]. However, adequate vitamin D levels were associated with preserved lung barrier function [15]. Moreover, vitamin D deficiency promotes protein catabolism in muscle tissue, which hinders recovery from the inflammatory process after a pathological event [16].

Various studies have shown that severe or fatal cases of COVID-19 disease are associated with elevated levels of urea, creatinine, tissue markers (lipid, kidney and muscle), C-reactive protein (CRP) and interleukin-6 (IL-6) and lower counts of lymphocytes (<1000/µL) and platelets (<100 × 109/L), as well as albumin [5,15,17,18,19]. These blood biomarkers are easy to determine and, in fact, are currently possible to determine in any laboratory in the hospital setting. They have been reported to be useful for monitoring and predicting disease outcome and prognosis [20,21]. A recently published study [22] created a model for individualized analysis of the risk of admission or death of patients with COVID-19 in an intensive care unit (ICU). This could be a useful tool for rapid clinical management and for achieving resilience of medical resources.

Following the above, the main objective of this study was to analyze the effects of vitamin D supplementation on muscle status in old patients after COVID-19 infection. Elders were selected because they represent a main risk population segment for COVID-19. In addition, aging is associated with muscle atrophy and impaired function, reflected by slowing of movements and weakness (known as sarcopenia). In this study, an assessment of circulating blood muscle markers was performed as a mechanism to observe the recovery process. Moreover, taking into account the respiratory distress caused by the virus infection, respiratory function (static and dynamic spirometry) and the impact on physical abilities were studied as an additional element of muscle function during the recovery process.

## 2. Materials and Methods

This pilot study was carried out with the collaboration and consent of patients of the Soria Norte Health Center. Participants suffered coronavirus infection and were diagnosed by PCR. These patients were evaluated and clinically surveyed at the Soria Norte Health Center and were treated and isolated during the period of infection. Potential volunteers were recruited over 3 weeks from the information provided at the primary care centers. Intervention started with those that met the inclusion criteria. Inclusion criteria were as follows: (a) healthy old male patients that have suffered from recent COVID-19 infection, and (b) no comorbidities such as cardiac, renal or respiratory pathologies or recent bone fractures. Male patients did not need to be registered in the emergency or medical services at the hospital. They underwent quarantine isolation at home under supervision of the hospital medical staff. As soon as the infection was known, vitamin D supplementation or placebo was provided to participants. Intervention was carried out in accordance with the Code of Ethics of the World Medical Association (Declaration of Helsinki) and approved by the Clinical Research Ethics Committee (CEIm) of the Valladolid East Health Area (PI 21-2104-NO HCUV).

A double-blind trial with 2 groups: placebo (PB) and treated with vitamin D (VD) (*n* = 15 per group), was carried out. The subjects were selected after attending a physician consultation and presenting symptoms compatible with COVID-19 infection. The PCR test was performed to confirm the infection. Details of the intervention were extensively explained to positive cases that expressed the intention to participate. The selected participants voluntarily signed an informed consent and started the treatment with oral solutions of either placebo or vitamin D. The study was performed as a double-blind trial: PB vs. VD (2000 IU/day). A basal blood analysis was performed at the moment of selection, corresponding to the first PCR positive test. Participants were homogenously distributed by the medical team with a similar *n* and no significant differences at the beginning in both groups (Table 1).

The two intervention groups (PB and VD) were treated for 6 weeks. VD was treated with a dose of 2000 IU/day. According to the scientific literature, this dose and time are adequate to reach stable circulating levels of vitamin D. We used vitamin D3 (cholecalciferol) Kern Pharma 2000 IU/mL oral solution in this study. Vitamin D was provided during food intakes because vitamin D is lipophilic and absorption is favored in the presence of fat.

### 2.1. Laboratory Determinations

Ten hours before blood extraction, a non-fat dinner was provided to participants. After a 30 min rest in the supine position at the hospital, 15 mL of blood sample was obtained at 8:30 a.m. from the antecubital vein. Participants were seated in a comfortable position using Vacutainer tubes. Blood was distributed in 2 tubes: 10 mL of blood was placed in one tube containing gel and clot activator to obtain the serum; additionally, 3–5 mL of blood was extracted and placed in an EDTA tube to analyze the hematological parameters. Serum was obtained by centrifugation at 2000 rpm for 15 min. Supernatant was extracted using a Pasteur pipette, transferred to a sterile tube and stored at −20 °C until the analysis.

Different parameters were measured by spectrophotometry in the hospital autoannalyzer (Hitachi 917, Boehringer Mannheim, Mannheim, Germany) for: total proteins, 25-OH vitamin D, creatinine, uric acid, creatine kinase (CK), aspartate aminotransferase (AST), alanine aminotransferase (ALT), lactate dehydrogenase (LDH), gamma-glutamyl transferase (GGT), aldolase (ALD) and CRP. The enzyme-linked fluorescent assay ELISA performed in a multiparametric analyzer (Minividas, Biomerieux, Marcy l’Etoile, France) was used to determine serum cortisol levels. ELISA was used to measure serum 25-OH vitamin D (DRG Instruments GmbH, Marburg/Lahn, Germany). All determinations were carried out in a certified hospital laboratory with the mandatory technical control. Finally, a hematological autoanalyzer (Sysmex XE2100, Sysmex, Kobe, Japan) was used for blood cell count and formula, hematocrit and hemoglobin determination.

### 2.2. Spirometry and Physical Status Determinations

Taking into account the respiratory involvement of these patients, we determined related functional parameters at the beginning, middle and end of the study by means of the relevant respiratory function tests (static and dynamic spirometry). From the functional point of view, we used FEV1 (forced expiratory volume) and respiratory flow by forced vital capacity (FVC). For these determinations, we used digital calibrated spirometers (Minispir Light, Medical International Research, Roma, Italy) that are transportable and easy to use.

In the physical test for functional assessment, we used the six-minute walk test (6MWT) (walking or running the maximum distance in this time) and strength in dominant hand by dynamometry. The 6MWT is a very widespread test in medicine, and we believe it is very useful in this case due to the age and physical condition of the participants.

A dietary and nutritional assessment was performed in order to rule out nutritional alterations and deficiencies.

### 2.3. Statistical Analysis

The sample size was arbitrarily estimated since it was a pilot study, and because the study was conducted according to the availability and voluntariness of the subjects. We had no dropouts, and the final sample size was 15 patients per group. To analyze the results, we used the SPSS statistical software. The results are presented as means + SEM. Statistical analysis was performed using the Mann–Whitney U test and the Friedman test for repeated samples, and comparisons of the results were carried out using the Wilcoxon paired test. The values were considered statistically significant at *p* < 0.05.

## 3. Results

The following Table 1, Table 2, Table 3 and Table 4 present the results obtained from patients who underwent COVID-19 infection. The data are presented in four columns: basal values of each group and after 6 weeks of treatment with vitamin D (2000 IU/day) or placebo.

Table 1 shows the data corresponding to the anthropometric characteristics of the studied patients. All of them were in a similar age range and had very similar physical characteristics.

Table 2 shows the data regarding hematological parameters, including red blood cells (RBC), white blood cells (WBC) and iron metabolism. Both ferritin and iron were in the healthy range. The rest of the parameters displayed no significant differences between groups, except for lymphocytes at the end of the intervention compared to the corresponding basal levels.

Regarding other circulating parameters (Table 3), we observed that the main parameters were in a healthy range, despite the virus infection. Nevertheless, CK (specific marker of muscle damage) decreased significantly after infection only in VD but not in PB compared to the corresponding basal values at the beginning of the intervention.

Table 4 shows the data regarding the mineral parameters modulated by vitamin D as well as circulating 25-OH vitamin D levels. Circulating levels of calcium, phosphorus and vitamin D displayed a significant increase only in the vitamin D-treated group compared to the basal values and to the PB group (only for calcium and vitamin D levels).

Physical condition and spirometry tests displayed a similar pattern in both groups studied (Table 5). We only observed slight increases in all parameters after vitamin D supplementation for 6 weeks, suggesting a tendency to improve in muscle and respiratory function recovery. However, the tests did not present statistically significant changes.

## 4. Discussion

The main obtained results indicate that serum 25-OH vitamin D levels increased, normalizing the mineral levels involved in phosphocalcic metabolism (Table 4). This result indicates a positive effect for vitamin D supplementation. However, focused on the objectives of this study, the effects of vitamin D supplementation on respiratory function and physical status (Table 5) in patients who have suffered coronavirus infection displayed no significant improvement. The same was observed for blood cell counts (Table 2), except for circulating markers of muscle damage such as CK (Table 3). In this context, circulating biomarkers represent quantitative determinations that reflect the pathophysiology of the disease, helping physicians to establish the severity of illness [22]. 

In our study, we did not observe consistent alterations in leukocyte counts as well as in the other hematological variables, such as RBC. These data coincide with those published by others [23]. Normal leukocyte counts were noticed in 70% of studied subjects (children infected by SARS-CoV-2). This result suggests that this is a first step to assure a correct immune response in the population studied in this report. In contrast, laboratory parameters in adults showed that increased leukocyte counts (mainly neutrophils) were common in patients with unfavorable COVID-19 progression [22,23]. These increases were a poor prognostic factor in the evolution of patients. In this context, lymphocytes play a key role in maintaining immune homeostasis and the inflammatory response to protect the body against viral infections [24,25]. One of the typical features of COVID-19 infection is a decrease in the number of lymphocytes and an increase in neutrophils [26]. In addition, minerals maintained normal levels during this study (Table 4), particularly iron (Table 2). This observation indicates no signs of anemia that could interfere with the anti-inflammatory response and the possible reinforcing effect of vitamin D in this process.

All these observations are reinforced by circulating CRP levels. CRP is an acute-phase protein considered as a biomarker of systemic inflammation and a significant contributor to disease pathogenesis. CRP is used to indicate infection or chronic inflammatory disease. In this context, inflammatory cytokines induce the production of CRP [27]. In the present study, we did not observe alterations in CRP, remaining at normal levels in the studied groups. Our data are in agreement with others published [28] that only observed a slight increase (3.6%), not a significant increase. The recent National Health and Nutrition Examination Survey (NHANES) supported a possible link between vitamin D levels and CRP in individuals with low 25-OH vitamin D levels (<21 ng/mL). The study concluded that inflammatory states appear to contribute to and worsen in subjects with vitamin D deficiency [29]. Elevated CRP may be a condition for unfavorable disease progression [30]. Although these tests provide an indication of the inflammatory process, the studies were mostly carried out in specific hospitalized populations. Therefore, further studies are needed to evaluate how these parameters progress in people with milder symptoms of inflammation. Taking these data into account, the results of the present report do not show significant differences between the studied groups, confirming that these were mild cases controlling COVID-19 infection.

On the other hand, the circulating levels of muscle damage markers (CK, AST, ALT and LDH) can be instrumental to monitor the muscle inflammatory process. Regarding AST, ALT and LDH, we did not observe significant changes as observed by others [31]. This occurred because certain markers such as AST and ALT are indicators of liver damage. In this context, around 75% of patients affected by COVID-19 have abnormal circulating levels of liver biomarkers, probably due to secondary hepatic damage. This could be the result of the systemic inflammatory response observed in this disease, and the use of hepatotoxic drugs in the management of patients with COVID-19 [31]. Additional research is necessary to confirm these observations.

On the other hand, we observed significant decreases in CK, the most specific muscle marker in the circulation. For this reason, CK is more reliable to determine acute muscle damage and subsequent recovery. Rhabdomyolysis is a common entity that often has a multifactorial etiology. The diagnosis of rhabdomyolysis is based mainly on elevated CK. An elevated plasma CK level is the most sensitive laboratory biomarker related to muscle injury [32,33]. Regarding COVID-19, immune-mediated muscle damage can be explained by the deposition of virus–antibody complexes in the muscles, viral toxins circulating in the blood, immunological cross-reactivity and virus-induced antigen expression in cell membranes [34]. According to the results of our report, we could hypothesize that vitamin D contributes to attenuate myositis, although additional research is required to confirm this point. In this context, we found a significant decrease in CK levels after 6 weeks of vitamin D treatment. This may suggest an immunomodulatory role of vitamin D in decreasing the muscle inflammatory process. Since the number of subjects in our study is small, the observed effects in the main portion of muscle markers have to be interpreted with caution. However, patients reported better sensations (subjective comment), likely corresponding to a decrease in the inflammatory process. Additional research is necessary to confirm all these points.

As we have communicated previously in a review, despite the evident benefit of vitamin D in muscle function, particularly in recovery from inflammation caused by exercise, it seems that musculoskeletal benefits occur when deficient or insufficient circulating levels of vitamin D (20–30 ng/mL) are restored [35]. At the end of our study, the levels of 25-OH vitamin D (VD group) were higher than 30 (31.32 ± 1.42). Therefore, it seems clear that it is crucial to preserve optimal vitamin D levels, both for maintaining basic body functions, and for muscle performance, recovery and anti-inflammatory control.

Regarding respiratory functional parameters and physical status condition, we only observed slight non-significant improvements in the tests performed, although patients reported feeling better. We interpret this as a subjective assessment derived both from the reduction in the inflammatory process and from the patients’ care, rather than from the treatment itself. In our study, we achieved an increase in serum levels of 25-OH vitamin D, which, according to the literature, could counteract the imbalance of some of the components of coronavirus infection and also manifest their own anti-inflammatory effects [35]. This speculative interpretation needs further research.

A point to take into account in future interventions is that new aspects such as the dose of vitamin D and the time of supplementation should be considered. Certain studies observed no effect of vitamin D supplementation (100,000 IU/month, for 3 months) on rehabilitation in patients with chronic obstructive pulmonary disease compared to placebo [36]. In the same vein [37], with a daily dose of 2000 IU of vitamin D for a period of 6 weeks (the same as our study), no changes were observed on physical performance scores. However, a meta-analysis [38] showed effects of vitamin D supplementation on muscle strength in the general adult population, except for participants with optimal circulating levels of 25 (OH)D. Altogether, the results are inconclusive, and additional research is necessary. 

The main limitation of this study is the low number of patients. Age might be an additional limitation to take into account. In this context, it is known that increasing age entails a reduction in muscle mass (sarcopenia) associated with a decrease in circulating levels of vitamin D [39]. Our study involved subjects in the age range of persons suffering from a certain degree of sarcopenia. This, added to the inflammatory process, could aggravate the clinical situation. Indeed, the results are not significant, but a tendency to improve was noticed. It seems that the most important effect of vitamin D supplementation on global muscle status is greater in participants with suboptimal vitamin D levels.

## 5. Conclusions

This report presents a route for muscle recovery after SARS-CoV-2 infection. The results suggest that vitamin D supplementation could be an alternative to consider in clinics for recovery of muscle damage caused by COVID-19 inflammation. Nevertheless, muscle recovery was observed after a mild infection process in a population of elders with a light degree of sarcopenia. One instrumental point for clinicians is that the recovery process can be monitored by using the information provided by current blood analysis. Although certain parameters display a tendency to improve, this could contribute to an increase in the patient’s quality of life in the long term. This point needs to be verified by additional research. Altogether, this is the first report that opens the possibility of vitamin D supplementation in clinics for post-COVID-19 recovery. 

## Figures and Tables

**Table 1 medicina-57-01079-t001:** Age and anthropometric characteristics of the studied participants.

Parameter	BASAL PB (*n* = 15)	BASAL VD (*n* = 15)
Age (years)	62.5 ± 1.5	60.6 ± 1.7
Weight (kg)	75.9 ± 5.6	81.1 ± 4.2
Height (cm)	170.6 ± 4.5	169.7 ± 4.7
BMI (kg/m^2^)	26.5 ± 5.1	27.9 ± 4.6

Abbreviations used: PB, placebo group; VD, vitamin D-supplemented group.

**Table 2 medicina-57-01079-t002:** Hematological and iron metabolism parameters.

Parameter	BASAL PB (*n* = 15)	BASAL VD (*n* = 15)	PB (*n* = 15)	VD (*n* = 15)
RBC (10^6^/µL)	4.9 ± 0.1	4.9 ± 0.1	4.9 ± 0.1	4.8 ± 0.1
Hematocrit (%)	44.6 ± 0.5	45.0 ± 0.6	44.8 ± 0.7	45.7 ± 0.7
Hemoglobin (g/dL)	14.6 ± 0.2	14.4 ± 0.3	14.5 ± 0.3	14.2 ± 0.2
Ferritin (ng/mL)	87.0 ± 16.7	80.9 ± 26.0	87.8 ± 19.7	80.1 ± 24.6
Fe (µg/dL)	95.9 ± 8.3	93.3 ± 7.1	105.0 ± 15.6	92.1 ± 6.1
WBC (10^3^/µL)	6.1 ± 0.3	5.9 ± 0.4	5.8 ± 0.4	4.8 ± 0.2
Neutrophils (%)	43.9 ± 3.3	45.1 ± 3.2	45.7 ± 3.4	47.3 ± 3.1
Lymphocytes (10^3^/µL)	2.7 ± 0.1	2.2 ± 0.2	2.4 ± 0.1 *	1.9 ± 0.1 *
Lymphocytes (%)	43.0 ± 3.1	42.7 ± 3.3	41.5 ± 3.1 *	39.2 ± 2.8 *
Monocytes (%)	7.1 ± 0.5	8.3 ± 0.4	7.0 ± 0.3	7.9 ± 0.6
Eosinophils (%)	2.5 ± 0.3	2.3 ± 0.3	2.4 ± 0.3	2.2 ± 0.2
Basophils (%)	0.9 ± 0.1	0.8 ± 0.2	0.8 ± 0.1	0.7 ± 0.1

* Indicates significant differences comparing the intervention groups (PB and VD) with respect to the corresponding basal values (Wilcoxon’s paired test). Abbreviations used: Fe, serum iron; PB, placebo group; RBC, red blood cells; VD, vitamin D-supplemented group; WBC, white blood cells.

**Table 3 medicina-57-01079-t003:** Circulating metabolites and tissue markers.

Parameter	BASAL PB (*n* = 15)	BASAL VD (*n* = 15)	PB (*n* = 15)	VD (*n* = 15)
Total proteins (g/dL)	7.1 ± 0.1	7.2 ± 0.1	7.2 ± 0.1	7.2 ± 0.1
Uric acid (mg/dL)	4.8 ± 0.4	4.9 ± 0.5	4.9 ± 0.3	5.3 ± 0.4
Creatinine (mg/dL)	1.3 ± 0.1	1.4 ± 0.2	1.2 ± 0.0	1.3 ± 0.1
AST (U/L)	34.3 ± 3.4	30.9 ± 2.2	31.3 ± 2.2	33.1 ± 2.5
ALT (U/L)	24.8 ± 1.6	24.8 ± 3.1	24.6 ± 1.3	25.6 ± 2.0
GGT (U/L)	18.9 ± 1.7	19.6 ± 1.4	18.6 ± 1.7	19.0 ± 2.5
CK (U/L)	145.9 ± 99.5	139.1 ± 93.2	158.6 ± 88.0	127.2 ± 85.1 *
LDH (U/L)	381.5 ± 18.7	379.7 ± 20.0	373.1 ± 20.5	386.2 ± 17.7
ALD (U/L)	7.0 ± 0.5	7.3 ± 0.4	7.1 ± 0.5	7.2 ± 0.7
CRP (mg/L)	2.3 ± 2.0	2.3 ± 2.4	3.1 ± 2.2	2.5 ± 1.9
Cortisol (µg/dL)	18.7 ± 1.2	18.8 ± 1.8	17.2 ± 1.3	19.8 ± 1.3

* Indicates significant differences comparing the intervention group VD with respect to the corresponding basal value (Wilcoxon’s paired test). Abbreviations used: ALD, aldolase; ALT, alanine aminotransferase; AST, aspartate aminotransferase; CK, creatine kinase; CRP, C-reactive protein; GGT, gamma-glutamyl transferase; LDH, lactate dehydrogenase; PB, placebo group; VD, vitamin D-supplemented group.

**Table 4 medicina-57-01079-t004:** Circulating minerals and 25-OH vitamin D levels.

Parameter	BASAL PB (*n* = 15)	BASAL VD (*n* = 15)	PB (*n* = 15)	VD (*n* = 15)
Ca (mg/dL)	10.0 ± 0.1	9.6 ± 0.1	9.5 ± 0.1	10.2 ± 0.1 *^†^
Mg (mg/dL)	2.1 ± 0.1	2.1 ± 0.1	2.0 ± 0.0	2.1 ± 0.1
P (mg/dL)	2.9 ± 0.1	2.7 ± 0.1	3.2 ± 0.0	3.1 ± 0.0 *
25-OH vitamin D (ng/mL)	21.2 ± 1.4	20.9 ± 1.8	19.4 ± 2.3	± 1.4 *^†^

* Indicates significant differences comparing VD with respect to the corresponding basal values (Wilcoxon’s paired test). ^†^ Indicates significant differences comparing the intervention groups (VD vs. PB) (Wilcoxon’s paired test). Abbreviations used: Ca, calcium; Mg, magnesium; P, phosphorous; PB, placebo group; VD, vitamin D-supplemented group.

**Table 5 medicina-57-01079-t005:** Spirometry and physical condition tests.

Test	BASAL PB (*n* = 15)	BASAL VD (*n* = 15)	PB (*n* = 15)	VD (*n* = 15)
6MWT (m)	834.3 ± 0.1	839.2 ± 0.4	829.1 ± 0.1	856.4 ± 0.1
Dynamometry (kg)	34.1 ± 2.3	35.6 ± 2.8	33.8 ± 3.0	36.1 ± 2.2
FVC (L)	3.9 ± 0.4	3.8 ± 0.1	3.9 ± 0.5	3.9 ± 0.6
FEVI (L/s)	3.0 ± 0.6	2.9 ± 0.8	2.9 ± 0.6	3.1 ± 0.6

Abbreviations used: FEV1, forced expiratory volume; FVC, forced vital capacity; PB, placebo group; 6MWT, six-minute walk test; VD, vitamin D-supplemented group.

## Data Availability

The data that support the findings of this study are available from the corresponding author, upon reasonable request.

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
