# Peer review of "Effect of Vitamin D Supplementation on Muscle Status in Old Patients Recovering from COVID-19 Infection"

_medicina, 2021, doi:10.3390/medicina57101079_

Round 1

Reviewer 1 Report

The paper by Caballero-García is current, but the study was conducted on a small group of patients.

Abstract

Indicate which form of vitamin D is being supplemented.

It is not correct that the serum levels of the biomarkers returned to the optimal values, only the CK.

Introduction

The high transmissibility and pathogenicity are referred to the virus (SARS-CoV-2), not to the disease (COVID-19).

The paragraph on cytokines is useless and should be either eliminated or extremely reduced; instead, the role of vitamin D in COVID-19 patients and its hypothetical protective role in this infection should be stressed (e.g. Gaudio et al. IJERPH 2021).

Outline the objectives well, avoiding interlocutory paragraphs such as: "The recovery process was documented by current blood markers ....... In this context, we hypothesize that hematological biomarkers can help ... "

Materials and methods

Have patients signed informed consent?

If the study was double blind, did placebo and vitamin D have the same formulation (oral solution)?

The authors write: “We used in the study, vitamin D3 Kern Pharma 2000 IU / ml oral solution and Cholecalciferol”. What does AND cholecalciferol mean?

Laboratory determinations

Have you determined serum levels of vitamin D? Or more correctly of 25-OH vitamin D? If so, please edit the entire text and tables to match.

Spirometry

Indicate the model of the spirometer used.

Results

In the text and in the abstract, the authors write several times that patients on vitamin D treatment felt better, but they do not refer to any questionnaire or scale.

Table 1 is essentially useless. It is clear that the age and height of the patients does not change after 6 months. Among other things, the changes in weight and BMI are not significant. It is sufficient to indicate the baseline values for both groups.

PC6m is not the acronym of the six minute walking test (maybe it is in Spanish), but the correct acronym is 6MWT.

In Table 5 indicate the unit of measurement (meters) for the data of the six minute walking test.

Discussion

“In our study, the levels of vitamin D (VD group) was higher than 30 (31.32 ± 1.42)”. Specify that the values were higher at the end of the study.

It is necessary to indicate the limits of the study - above all, that the study was conducted only in a small number of patients.

The English language should be thoroughly revised by a professional editor.

Author Response

Reviewer 1 (round 1)

The paper by Caballero-García is current, but the study was conducted on a small group of patients.

ANSWER: This is a pilot study. For this reason the n is small. We have indicated this point in the abstract and in the first sentence of Materials and Methods.

Abstract

Indicate which form of vitamin D is being supplemented.

ANSWER: The form of vitamin D is cholecalciferol (vitamin D3). This is now indicated in the Abstract.

It is not correct that the serum levels of the biomarkers returned to the optimal values, only the CK.

ANSWER: Circulating tissue markers have different half-lives, being CK shorter (around 1 day) than amino-transferases (around 3-6 days). We have changed the sentence accordingly.

Introduction

The high transmissibility and pathogenicity are referred to the virus (SARS-CoV-2), not to the disease (COVID-19).

ANSWER: This has been changed according to Reviewer suggestion.

The paragraph on cytokines is useless and should be either eliminated or extremely reduced; instead, the role of vitamin D in COVID-19 patients and its hypothetical protective role in this infection should be stressed (e.g. Gaudio et al. IJERPH 2021).

ANSWER: This explanation is to put in context the cytokine storm which is characteristic of COVID-19. We have reduced the content of this paragraph according to Reviewer suggestions. On the other hand, the debate regarding low levels of vitamin D and progression of COVID has been included in the Introduction. The new Reference 14 (Gaudio et al, 2021) has been cited.

Outline the objectives well, avoiding interlocutory paragraphs such as: "The recovery process was documented by current blood markers ....... In this context, we hypothesize that hematological biomarkers can help ... "

ANSWER: We wanted to emphasize that the current blood analysis could be an instrumental tool to follow the course of the disease. In any case, we have focused on the objectives and delete interlocutory sentences in the last paragraph of the Introduction.

Materials and methods

Have patients signed informed consent?

ANSWER: Yes, patients signed an informed consent. This is now indicated in the second paragraph of Materials and Methods.

If the study was double blind, did placebo and vitamin D have the same formulation (oral solution)?

ANSWER: Yes, both are oral solutions. This is now indicated in the second paragraph of Materials and Methods.

The authors write: “We used in the study, vitamin D3 Kern Pharma 2000 IU / ml oral solution and Cholecalciferol”. What does AND cholecalciferol mean?

ANSWER: This is a typing mistake. This has been corrected in the third paragraph of Materials and Methods.

Laboratory determinations

Have you determined serum levels of vitamin D? Or more correctly of 25-OH vitamin D? If so, please edit the entire text and tables to match.

ANSWER: Yes, 25-OH vitamin D was determined. We have changed in several parts of the manuscript, including Table 4.

Spirometry

Indicate the model of the spirometer used.

ANSWER: The model of spirometers was indicated. See first paragraph of Section 2.2.

Results

In the text and in the abstract, the authors write several times that patients on vitamin D treatment felt better, but they do not refer to any questionnaire or scale.

ANSWER: As we explained in paragraphs 6 and 8 of Discussion, this is a subjective assessment made by patients. We gave this information in the Discussion section but not in the Results, because we used no scales or questionnaires. However, this is an information that opens a new line for future research to complete this study, as we indicated in paragraph 8 of Discussion

Table 1 is essentially useless. It is clear that the age and height of the patients does not change after 6 months. Among other things, the changes in weight and BMI are not significant. It is sufficient to indicate the baseline values for both groups.

ANSWER: Only baseline values were indicated as suggested by the Reviewer. See new version of Table 1.

PC6m is not the acronym of the six minute walking test (maybe it is in Spanish), but the correct acronym is 6MWT.

ANSWER: The acronym has been changed accordingly.

In Table 5 indicate the unit of measurement (meters) for the data of the six minute walking test.

ANSWER: The unit (m) has been indicated.

Discussion

“In our study, the levels of vitamin D (VD group) was higher than 30 (31.32 ± 1.42)”. Specify that the values were higher at the end of the study.

ANSWER: The sentence was changed according to reviewer suggestion (see paragraph 7 of Discussion).

It is necessary to indicate the limits of the study - above all, that the study was conducted only in a small number of patients.

ANSWER: Limitations of the study (n and age) have been indicated in the last paragraph of Discussion.

The English language should be thoroughly revised by a professor of the university.

Reviewer 2 Report

This paper presents the results of  analyzing the effect of vitamin D in muscular recovery in patients after infection with SARS-CoV-2A (COVID-19). Even new information is identified, however, the findings is weakened by several issues including incomplete description of background, method and results, inconsistency of gap identified, results and discussion, and issues in the logical presentation of the study aims.  Details relating to these and other issues are presented below.

Title and abstract:

  1. Suggest: revise the title and abstract part: Both does not match research purpose and content of main parts of the draft.
  2. Suggest to revise the title: Title does not match research purpose and content: (after considering the focus of this study). Suggest to integrate the method of this study...

Introduction

  1. Authors should revise introduction section: appropriate paragraphs using topic sentences, which are relevant to the title. Each paragraph should first start with a topic sentence to introduce an overall theme of the paragraph, followed by supporting sentences to support the topic sentence.
  2. The current logical flow of the background makes it difficult to understand what the significance/importance of this particular topic is and why this study should be conducted. The background should be revised to showcase why this topic is important, what the current findings present/limitations are, and therefore what this study aims to show.
  3. Paragraphs should be re-formatted to improve fluency:
  4. Suggest to reduce the number of paragraph: They should be broken by major topics.
  5. The authors should check whether all sentences are properly cited.
  6. Some of the references seems not to be relevant.
  7. Line 92: Suggest to revise “authors to establish a predictive”
  8. Line 100: Suggest to revise “In addition and to taking into account the”

Methods

  1. Suggest to revise the methodology

- Authors should include more articles related to the causal relation or mechanism between intervention and muscle recovery.

- provide how to determine the size of sample.

Discussion  

  1. Here again, the current logical flow of the result section makes it difficult to understand what the significance/importance of this particular topic.
  • It is useful to focus on one idea or discussion point per paragraph and avoid discussing only one study per paragraph.
  • Synthetize information obtained from several studies by critically reviewing one common topic of several articles.

Conclusion

  1. Suggest to revise the conclusion

-  Clearly state the answer to the main research question.

- Summarize and reflect on the research.

- suggest recommendations for future work on the topic (after considering the focus of this study).

- Indication what new knowledge this study has contributed.

(end)

Author Response

Reviewer 2 (round 1)

This paper presents the results of analyzing the effect of vitamin D in muscular recovery in patients after infection with SARS-CoV-2A (COVID-19). Even new information is identified, however, the findings is weakened by several issues including incomplete description of background, method and results, inconsistency of gap identified, results and discussion, and issues in the logical presentation of the study aims.  Details relating to these and other issues are presented below.

Title and abstract:

  1. Suggest: revise the title and abstract part: Both does not match research purpose and content of main parts of the draft.
  2. Suggest to revise the title: Title does not match research purpose and content: (after considering the focus of this study). Suggest to integrate the method of this study...

ANSWER: We have matched the research purpose indicating in the title and in the Abstract, that participants were old and in the phase of recovery after SARS-CoV-2A infection. We wanted to check the status of the muscle tissue, because this population is prone to undergo sarcopenia This a relevant aspect that we have indicated in the Discussion (last paragraph).

Introduction

  1. Authors should revise introduction section: appropriate paragraphs using topic sentences, which are relevant to the title. Each paragraph should first start with a topic sentence to introduce an overall theme of the paragraph, followed by supporting sentences to support the topic sentence.
  2. The current logical flow of the background makes it difficult to understand what the significance/importance of this particular topic is and why this study should be conducted. The background should be revised to showcase why this topic is important, what the current findings present/limitations are, and therefore what this study aims to show.
  3. Paragraphs should be re-formatted to improve fluency:
  4. Suggest to reduce the number of paragraph: They should be broken by major topics.
  5. The authors should check whether all sentences are properly cited.
  6. Some of the references seems not to be relevant.

ANSWER: We have taken into account this list of recommendations to organize the Introduction. We have reduced the number of paragraphs that are focused in main topics related to the background. These are distributed as follows:

  • Paragraph 1: General introduction to the disease, indicating the main symptoms.
  • Paragraph 2: The role of cytokines in the inflammatory process that occurs in COVID, known as “cytokine storm”. This a key point, because vitamin D is an immunomodulatory micronutrient (Paragraph 3).
  • Paragraph 3: The role of vitamin D in the progression of the disease. Vitamin D seems to exert a modest effect on the respiratory syndrome, meanwhile it could work better in preserving muscle status. This is the aim of the study and indiated in the paragraph accordingly.
  • Paragraph 4: The importance of circulating parameters determined by current blood analysis. This is an important message for health professionals, because these parameters can give an idea regarding the progression of the disease in an easy way. Subsequent more specific determinations can be performed, but the current blood analysis is the first approach. In this context, clinicians are the main readers of Medicina and this information could be instrumental for them. In addition, current blood analysis is the main methodology used in the present study.
  • Paragraph 5: Objective of the study.

  1. Line 92: Suggest to revise “authors to establish a predictive”

ANSWER: This a main conclusion raised in Reference 22: The use of current blood analysis parameters helped to the authors to establish a predictive model of progression of the disease. We are just citing the main conclusion of this work that reinforces the Methodology used in our manuscript.

  1. Line 100: Suggest to revise “In addition and to taking into account the”

ANSWER: The sentence has been changed accordingly.

Methods

  1. Suggest to revise the methodology

- Authors should include more articles related to the causal relation or mechanism between intervention and muscle recovery.

ANSWER: This is the first study describing the effect of vitamin D supplementation in the maintaining muscle status in patients recovering from COVID infection. There are not articles related to this particular topic.

- provide how to determine the size of sample.

ANSWER: The sample size was arbitrarily estimated because it was a pilot study, and because the study was conducted according to the availability and voluntariness of the subjects.

Mr reviewer, please bear in mind that the study was carried out in a small city (Soria), in which, at the beginning, COVID-19 only affected people of a high age like the subjects included in this study. This meant that a mathematical sample selection could not be carried out. This meant that the study included volunteers who wanted to participate, and in this place (Soria) there are few possibilities for other types of selection, which is why we depend on the recruitment of these patients.

Discussion  

  1. Here again, the current logical flow of the result section makes it difficult to understand what the significance/importance of this particular topic.
  • It is useful to focus on one idea or discussion point per paragraph and avoid discussing only one study per paragraph.
  • Synthetize information obtained from several studies by critically reviewing one common topic of several articles.

ANSWER: We avoided to cite particular studies and focused in general findings by several studies, except in certain cases that we found relevant to discuss separately. Taking into account this list of recommendations, we have organized the Discussion following the flow of Results:

  • Paragraph 1: Main observations of the study.
  • Paragraph 2: Discussion of results from Table 2. These are key indicators of the status of the immune system and inflammation. Among others, we commented regarding iron status. Iron deficiency is related to anemia and this could impair the anti-inflammatory response and the reinforcing role of vitamin D.
  • Paragraph 3: Discussion of results from Table 3, focusing first in CRP as a main marker of inflammation and connecting to the data discussed previously (Paragraph 2).
  • Paragraphs 4 and 5: Continuation of discussion of results from Table 3 focusing now in tissue biomarkers, mainly in CK. Due to the short half-live in circulation, CK is the most reliable marker to follow the progression of muscle damage and subsequent recovery. Finally, we connect the recovery of muscle damage with vitamin D supplementation, although we suggested being cautious with interpretation of these results.
  • Paragraph 6: The main conclusions regarding correct circulating vitamin D levels are presented.
  • Paragraph 7: Discussion of results regarding spirometry and physical condition tests. No conclusive results were obtained. We present some interpretations that need to be confirmed in a future research.
  • Paragraph 8: Proposal for future research. We present several references where results present certain variability. A as first step, we suggest that it is necessary to establish the dose and length for vitamin D supplementation.
  • Paragraph 9: Limitations of the study, focusing mainly in the n and age of participants. In this context, participants suffer light degree of sarcopenia and this could determine the recovery process. This is a point to consider in future research.

Conclusion

  1. Suggest to revise the conclusion

-  Clearly state the answer to the main research question.

- Summarize and reflect on the research.

- Suggest recommendations for future work on the topic (after considering the focus of this study).

- Indication what new knowledge this study has contributed.

ANSWER: Conclusions have been changed accordingly.

Round 2

Reviewer 1 Report

The manuscript has been improved by the authors, modifying it according to the previous suggestions.

Author Response

Thank you very much

Reviewer 2 Report

Thank you for addressing the comments. Still, there are several points that needs to be addressed as stated below.

Introduction

Line 83: authors should state why muscle status in old patients?

  1. Material and methods

Suggest to provide:

1) the inclusion criteria of subject;

2) time frame of the intervention and collecting specimen from the respondents.

Line 214: , and focused on the objectives of the study, the effects of vitamin D supplementation on respiratory function and 215 physical status (Table 5)=> focused on the objectives of the study, the effects of vitamin D supplementation on respiratory function and 215 physical status (Table 5)

Line 220; helping physicians to stablish the severity of illness => helping physicians to establish the severity of illness

Line 245: Elevated CRP may condition unfavorable disease progressionè Elevated CRP may be condition unfavorable disease progression

Line 261-262: we observed significant decreases in CK, the muscle marker that 261 displays the shorter half-live in circulationè?

Line 275: “However, patients reported better sensations (subjective com- 275 ment), corresponding likely to a decrease in the inflammatory process”: is it in the result?

Line 289-291: In our study, we have achieved an increase in serum levels of 25-OH vitamin D, which, according to the literature, these levels counteract the imbalance of some of the components of coronavirus infection and also manifest their own anti-inflammatory effects. This speculative interpretation needs a further research.==> reference?

Line 311: edit: “This report presents a way for muscle recovery caused by SARS-CoV-2 infection.”

Author Response

REVIEWER-2 (round 2)

Thank you for addressing the comments. Still, there are several points that need to be addressed as stated below.

ANSWER: We thank again the comments made by the Reviewer that have improved considerably the manuscript.

Introduction

Line 83: authors should state why muscle status in old patients?

ANSWER: Aging is related with muscle atrophy and impaired function reflected by slowing of movements and weakness. In addition, elders are target population segment for COVID. Both statements are indicated now in the corrected version of the manuscript.

Material and methods

Suggest to provide:

1) The inclusion criteria of subjects;

ANSWER: The inclusion criteria are the following: a) Healthy old people that have suffered from a recent COVID, b) No comorbidities, such as cardiac, renal or respiratory associated pathologies or recent bone fractures. Criteria are indicated in the corrected version of the manuscript.

2) Time frame of the intervention and collecting specimen from the respondents.

ANSWER: Potential volunteers to participate in the study were recruited during 3 weeks from the information provided at the Primary Care Centers. Then, we started the intervention with those that meet the inclusion criteria. This is indicated in the corrected version of the manuscript.

Line 214: , and focused on the objectives of the study, the effects of vitamin D supplementation on respiratory function and 215 physical status (Table 5)=> focused on the objectives of the study, the effects of vitamin D supplementation on respiratory function and 215 physical status (Table 5).

ANSWER: This typo error has been corrected in the manuscript.

Line 220; helping physicians to stablish the severity of illness => helping physicians to establish the severity of illness

ANSWER: This typo error has been corrected in the manuscript.

Line 245: Elevated CRP may condition unfavorable disease progression => Elevated CRP may be condition unfavorable disease progression.

ANSWER: This typo error has been corrected in the manuscript.

Line 261-262: we observed significant decreases in CK, the muscle marker that 261 displays the shorter half-live in circulation =>?

ANSWER: We guess that we have to indicate that CK is the most specific muscle marker in circulation compared to AST and ALT that indicate liver damage at the same time. This is now reflected in the new version of the manuscript.

Line 275: “However, patients reported better sensations (subjective com- 275 ment), corresponding likely to a decrease in the inflammatory process”: is it in the result?

ANSWER: This is not a result, just a general comment made by patients supplemented with vitamin D. We need to determine with a specific test this statement in a future research. This is why we did not consider this comment as a result, but this statement in the Discussion opens the possibility for a further research that readers can follow, as indicated in the manuscript.

Line 289-291: In our study, we have achieved an increase in serum levels of 25-OH vitamin D, which, according to the literature, these levels counteract the imbalance of some of the components of coronavirus infection and also manifest their own anti-inflammatory effects. This speculative interpretation needs a further research.==> reference?

ANSWER: Reference 35 reports the role of vitamin D in the control of inflammatory processes, mainly in the field of exercise and physical activity. This prompted us to speculate that vitamin D could counteract the harmful effects of COVID. However, this is only a hypothesis that need further research, as indicated in the manuscript.

Line 311: edit: “This report presents a way for muscle recovery caused by SARS-CoV-2 infection.”

ANSWER: The new correct sentence is: “This report presents a way for muscle recovery after SARS-CoV-2 infection.”